# Acute Responses of Core Muscle Activity during Bridge Exercises on the Floor vs. the Suspension System

**DOI:** 10.3390/ijerph18115908

**Published:** 2021-05-31

**Authors:** Jim T. C. Luk, Freeman K. C. Kwok, Indy M. K. Ho, Del P. Wong

**Affiliations:** 1Department of Sports and Recreation, Technological and Higher Education Institute of Hong Kong (THEi), Hong Kong; freemankkc@gmail.com (F.K.C.K.); indyho@thei.edu.hk (I.M.K.H.); 2School of Nursing and Health Studies, The Open University of Hong Kong, Hong Kong; delwong.cuhk@gmail.com; 3Titi Sports Technology, Shenzhen 518000, China

**Keywords:** surface EMG, prone bridge, supine bridge, unstable surface, plank

## Abstract

This study aimed to compare the neuromuscular activation of selected core musculature in supine and prone bridge exercises under stable versus suspended conditions. Forty-three healthy male participants were recruited to measure the electromyographic activities of the rectus abdominis (RA), lumbar multifidus (LM), thoracic erector spinae (TES), rectus femoris (RF), gluteus maximus (GM), and biceps femoris (BF) during supine and prone bridge exercises under six conditions: control, both arms and feet on the floor (Prone_con_ and Supine_con_); arms on the floor and feet on the suspension system (Prone-Feet_suspension_ and Supine-Feet_suspension_); and arms on the suspension system and feet on the floor (Prone-Arm_suspension_ and Supine-Arm_suspension_). Prone-Arm_suspension_ yielded significantly higher activities in the RA, RF, TES, and LM than Prone-Feet_suspension_ (*p* < 0.01) and Prone_con_ (*p* < 0.001). Moreover, Supine-Feet_suspension_ elicited significantly higher activities in the RA, RF, TES, LM, and BF than Supine-Arm_suspension_ (*p* < 0.01) and Supine_con_ (*p* < 0.001). Furthermore, Supine-Feet_suspension_ elicited significantly higher activities in the RF, TES, and BF than Supine_con_ (*p* < 0.01). Therefore, if the RA and/or RF were the target training muscles, then Prone-Arm_suspension_ was recommended. However, if the TES, LM, and/or BF were the target training muscles, then Supine-Feet_suspension_ was recommended.

## 1. Introduction

In recent years, the importance of core stability and strength is widely recognized in rehabilitation and athletic training [1]. The core muscles serve as the center of the functional kinetic chain, contribute to resistance against spinal perturbations [2], and transfer power to the terminal segments in athletic activities [3]. In the rehabilitative field, core-strengthening exercises can decrease the risk of injuries by increasing muscle power and endurance [4]. A meta-analysis reveals that core stability exercises are better than general exercises in decreasing pain and restoring physical functions in patients with chronic lower back pain [5]. Another previous study has found that the training of trunk-stabilizing muscles plays an important role in preventing lower extremity injuries in the athletic population [6], which has a direct impact on the mobility of limbs [7] and performance-related fitness [8,9].

Typical strengthening exercises for core muscles include supine bridge, prone bridge, and crunch exercises [10]. Performing exercises on unstable surfaces may be more effective than performing on stable surfaces in increasing the somatosensory feedback and enabling continuous adjustment of the overall position [11]. Common destabilizing strategies include the use of swiss balls, BOSU balls, and suspension straps. In particular, previous studies have found that performing dynamic resistance exercises on unstable surfaces generally reduce the peak force, rate of force development, and agonist muscle activity compared with performing on stable surfaces [12]. In contrary to dynamic exercises, performing static isometric exercises on unstable surfaces increases the muscle activity of trunk-stabilizing muscles. Previous studies have found that performing prone bridge exercises on a swiss ball or with hands on suspension straps results in higher muscle activity in the rectus abdominis (RA) compared with performing the same exercise on the floor [11,13]. Likewise, performing prone and supine bridges on a whole-body vibration platform induces higher trunk muscle activity compared with performing the same exercise on the floor [14]. However, research on the muscle activity during prone bridge exercises with feet on the suspension straps is limited [15], although previous studies have reported the use of suspension training in athletic populations, such as synchronized swimming [16] and Chinese boxing [17], and in clinical patients with brain injury [18] and functional training in older adults [19].

During suspension training, the participant usually suspends from the handles of the straps by either their hands or feet, while the non-suspended pair of extremities is in contact with the ground [20]. A recent systematic review has found that previous studies of suspension training exercises primarily focus on dynamic exercises (such as push-ups, inverted row, and hamstring curl) and static exercises (such as prone bridge) [21]. Most of the studies only focus on the core muscle activities during supine bridge exercises [22] or prone bridge-related exercises [2,11,15,23,24], but a direct comparison between supine and prone bridges with and without the use of a suspension system is still lacking. In addition, literature comparing the muscle activities between prone and supine bridges performed using a suspension system on arms versus floor exercises [20] or using a swiss ball as an unstable surface [25] is limited. The comparison of muscle activities between stable and unstable surfaces is better with similar posture and trunk inclination; however, most of the literature compares the exercises with very different trunk inclination because of the use of swiss balls or improper length of suspension straps [11,26,27]. Moreover, to date, no study has directly investigated the muscle activity when performing the two common static core muscle training exercises (i.e., prone and supine bridges), particularly when either the hands or feet are on the suspension straps.

Therefore, this study aimed to compare the neuromuscular activation of selected core musculature in supine and prone bridge exercises under stable versus unstable (i.e., suspension system) conditions. This study provided empirical evidence for trainers and therapists to make an informed decision when selecting between the two bridge exercises (i.e., prone and supine bridges) in three different forms to maximize training effectiveness.

## 2. Materials and Methods

### 2.1. Experimental Approach to the Problem

Within-participant repeated-measure design was used to compare the muscle activities under six conditions: control, both arms and feet on the floor (Prone_con_ and Supine_con_); arms on the floor and feet on the suspension system (Prone-Feet_suspension_ and Supine-Feet_suspension_); and arms on the suspension system and feet on the floor (Prone-Arm_suspension_ and Supine-Arm_suspension_). During all exercises, surface electromyography (sEMG) of the rectus abdominis (RA), lumbar multifidus (LM), thoracic erector spinae (TES), rectus femoris (RF), gluteus maximus (GM), and biceps femoris (BF) was measured. Exercise sequence was counterbalanced. EMG data were normalized with maximal voluntary contraction (MVC) values before being used for comparison among different exercise conditions.

### 2.2. Participants

Forty-three healthy male subjects (age: 21.40 ± 1.78 years; height: 174.74 ± 7.68 cm; mass: 64.42 ± 8.25 kg; body fat: 10.94% ± 3.20%, abdominal skinfold thickness: 12.11 ± 5.31 mm) volunteered for this study after providing a written informed consent and Physical Activity Readiness Questionnaire. The inclusion criteria for participants were as follows: (1) no history of lower back pain for more than 6 weeks before the study, (2) no surgery or trauma of the abdominal/back area requiring stitches within the last 12 months, (3) no lumbopelvic pain during an active straight leg raise test, (4) no muscle pain or discomfort when performing the selected exercises in this study, (5) with previous experience or background of training with suspension system and (6) abdominal skinfold measurement less than 25 mm and body fat less than 24% because percentages greater than this cut-off increased impedance of the EMG signal [28,29]. On the day of testing, participants reported having abstained from caffeine and vigorous exercise in the previous 12 h. This study was approved by the Research Ethics Committee of the Technology and Higher Educational Institution (Reference No: THEi-ILO/5/1/8) and in accordance with the Declaration of Helsinki [30].

### 2.3. Procedures

Each subject completed the anthropometric measures and screening protocol conducted by a registered physiotherapist. Prior to electrode placement, each subject performed a 5 min ergometer warm-up, and the subjects were familiarized with the exercise requirements to ensure that they could perform all selected exercises using a proper form.

Bipolar silver/silver-chloride electrodes (Ambu BlueSensor T, Ambu, Copenhagen, Denmark) with an inter-electrode distance of 30 mm were applied to bilateral RA, LM, TES, RF, GM, and BF and aligned with the pennate muscles on the bellies. Placements of electrodes for the RA [10], LM [31], TES [31], RF [32], GM [33], and BF [34] were reported in previous studies. The skin preparation and identification of electrode locations were performed following the procedure from the literature [33].

The MVC of each muscle was measured in different body positions as reported in previous studies: RA [35], RF [36], LM, TES, GM, and BF [37]. Three trials of 5 s MVC were performed for each muscle with at least 50 s of rest between trials [10]. The middle 3 s averaged root mean square (RMS) was collected. The greatest RMS among the three MVC trials was used to normalize each of the corresponding muscle activities in the subsequent tests, which allowed inter-individual comparison with the individual maximum [38].

Upon completion of MVC exercises, the six bridge exercises were performed in a counterbalanced order. Each variation of the bridge exercises was held for a 30 s isometric contraction and was repeated for two trials with a 5 min rest interval. The averaged RMS of the electrical muscle activities from the 12.5th to the 17.5th second was collected. The arms of the subjects were kept constantly perpendicular to ground level in prone bridge conditions to ensure internal consistency of the technique, whereas 90° knee flexion was maintained in supine bridge conditions (Figure 1 and Figure 2). The suspension system (TRX, Fitness Anywhere, Inc., San Francisco, CA, USA) was anchored to a metal frame at 4 m from the ceiling. Handles were placed approximately 6–8 cm above the ground for Prone-Feet_suspension_, Prone-Arm_suspension_, and Supine-Feet_suspension_. The length of the suspension straps was adjusted for each subject in Supine-Arm_suspension_ to ensure that the head of the subject was approximately 6–8 cm above the ground.

The proper techniques of each exercise were as follows: regular prone bridge (Prone_con_) is a prone bridge position on an exercise mat with arms held perpendicular to ground level. Only the palms and toes were in contact with the ground. Subjects were instructed to maintain a neutral head and spine with extended leg position throughout the exercise (Figure 1a). Prone bridge with arm suspension (Prone-Arm_suspension_) is similar to the regular prone bridge position with feet placed together on the ground, while the hands were placed inside the suspension straps with a neutral grip position and straight-arm position perpendicular to ground level (Figure 1b). Prone bridge with feet on a suspension system (Prone-Feet_suspension_) is similar to the regular prone bridge position with palms on the ground, while the instep was placed on the suspension straps (Figure 1c). Regular supine bridge (Supine_con_) is a supine bridge position with feet shoulder-width apart. Arms were placed beside the torso, and the knees were flexed at 90° with both feet resting on the exercise mat. The pelvis was lifted and aligned with the thigh (Figure 2a). Supine bridge with arm suspension (Supine-Arm_suspension_) is similar to the regular supine bridge position, but the suspension handle was held in a neutral grip position and straight-arm position perpendicular to ground level (Figure 2b). The supine bridge with feet on a suspension system (Supine-Feet_suspension_) is similar to the regular supine bridge position, but their feet were placed onto the suspension strap shoulder width apart (Figure 2c).

sEMG was measured at a sampling rate of 1500 Hz using a 12-channel data logger allied to the wireless TeleMyo Desktop Direst Transmission System (Noraxon, Inc., Arizona, USA). All collected signals were subsequently band-pass filtered between 10 and 500 Hz and then rectified and smoothed by calculating the RMS with a 50 ms sliding window. In MVC, the greatest RMS calculated from each trial of each side was averaged into one value to represent the muscle group bilaterally. In the six exercises, the greatest RMS calculated from each trial of each side was averaged and normalized, which was expressed as %MVC. The four levels of %MVC developed in previous studies were adopted to compare the muscle activation level among different conditions, with <21% as low, 21–40% as moderate, 41–60% as high, and >60% as very high [4,39].

### 2.4. Statistical Analysis

Data were reported as means and SD values. An intraclass correlation coefficient (ICC) with absolute agreement was used to determine the same-day test–retest reliability of sEMG recordings [40]. The classification of ICC values was as follows: <0.50 poor reliability, 0.50–0.75 moderate reliability, 0.76–0.90 good reliability, and >0.90 excellent reliability [41]. One-way repeated-measure analysis of variance was conducted to determine the global difference in sEMG activity among the six conditions. Post hoc pairwise comparisons with Bonferroni corrections were used. Statistical significance was set at α = 0.05. Cohen’s *d* effect size (ES) values were also determined with the following cut-off values: <0.40 small, 0.40–0.70 moderate, and >0.7 large [42].

## 3. Results

Table 1 shows the test–retest reliability of EMG measurements between the two repeated trials. Among the 36 sets of reliabilities, 8, 24, and 4 had excellent, good, and moderate ICC reliabilities, respectively.

Significant differences were found among the three conditions of prone exercises in the RA (F = 118.9, *p* < 0.001; Table 2 and Figure 3), RF (F = 153.8, *p* < 0.001), TES (F = 258.2, *p* < 0.001), LM (F = 251.0, *p* < 0.001), GM (F = 128.8, *p* < 0.010), and BF (F = 172.9, *p* < 0.001). The post hoc results indicated that Prone-Arm_suspension_ yielded significantly higher activities in the RA, RF, TES, and LM as compared with Prone-Feet_suspension_ (*p* < 0.010, all large ES: RA = 3.64, RF = 1.01, TES = 1.30, and LM = 1.71) and Prone_con_ (*p* < 0.001, all large ES: RA = 7.28, RF = 2.15, TES = 3.29, and LM = 3.35). In addition, Prone-Feet_suspension_ yielded significantly higher activities in the RA, RF, TES, and LM as compared with Prone_con_ (*p* < 0.001, all large ES: RA = 1.42, RF = 0.78, TES = 1.26, and LM = 1.00). Post hoc and ES evaluation revealed that GM and BF activations during Prone-Arm_suspension_ were significantly greater than those during Prone_con_ (*p* < 0.010, ES: GM = 0.53 and BF = 0.98).

Regarding the three supine conditions, significantly different muscle activations were found among the three conditions for all muscles (Table 2 and Figure 3): RA (F = 118.9, *p* < 0.001), RF (F = 153.8, *p* < 0.001), TES (F = 258.2, *p* < 0.001), LM (F = 251.0, *p* < 0.001), GM (F = 128.8, *p* < 0.010), and BF (F = 172.9, *p* < 0.001). Post hoc analysis revealed that Supine-Feet_suspension_ elicited higher muscle activities in the RA, RF, TES, LM, and BF as compared with Supine-Arm_suspension_ (*p* < 0.010, ES: RA = 1.46, RF = 0.69, TES = 0.78, LM = 0.43, and BF = 2.59) and Supine_con_ (*p* < 0.001, ES: RA = 1.17, RF = 1.08, TES = 1.03, LM = 0.45, and BF = 3.25). Activations of the RF, TES, and BF were also significantly higher during Supine-Arm_suspension_ in comparison with Supine_con_ (*p* < 0.010, ES: RF = 0.27, TES = 0.28, and BF = 0.49). However, Supine-Feet_suspension_ elicited significantly lower muscle activity for the GM when compared with Supine-Arm_suspension_ (*p* < 0.001, ES = 1.06) and Supine_con_ (*p* < 0.001, ES = 1.31).

## 4. Discussion

This study aimed to compare the neuromuscular activation of the selected core musculature in supine and prone bridge exercises under stable versus unstable (i.e., suspension system) conditions. The results of this study indicated that the instability provided by the suspension system increased the activation levels of core musculatures, such as the RA, LM, TES, RF, and BF, during static exercises, such as supine and prone bridges. This study was the first to show that the use of suspension training increased muscle activation as compared with conventional supine/prone bridge exercises performed on a stable surface. In particular, a high level of activations of the anterior musculature (i.e., RA) in prone bridge and posterior musculature (i.e., TES, LM, and BF) in supine bridge was observed.

Although TES, LM, GM, and BF remained in the low activation level in prone bridge exercises in this study, they demonstrated a significant difference in muscle activities across different exercise conditions. The result was congruent to that of the previous study, showing significantly greater activation of TES in Prone-Arm_suspension_ when compared with Prone-Feet_suspension_ and Prone_con_ [2]. However, this result was inconsistent with that of a previous study showing no difference in the activation of TES and GM when performing the prone bridge exercise with and without foot suspension [20]. Nonetheless, prone bridge exercises should not be considered as effective training for TES, LM, GM, and BF because of the low activation level of these muscles.

The current results showed that the highest muscle activities of the RA and RF were found in Prone-Arm_suspension_, followed by Prone-Feet_suspension_ and Prone_con_ (Figure 3). This result was consistent with that of previous studies showing that the activation level of the RA ranged from moderate [2,10] to low [43] in Prone_con_. The effect of suspension training on RA activation in our study was consistent with that of the previous similar studies. Previous studies indicated a 33% increase in RA muscle activation with Prone-Feet_suspension_ [23]. The RA activation level is significantly very high (91%) in Prone-Arm_suspension_, high (55%) in Prone-Feet_suspension_, and moderate (36%) in Prone_con_ [2]. Furthermore, a significant increase in the muscle activation of the RA, RF, and TES was found when push-up was performed on a suspension training system [22]. Based on the findings of the present study, moderate muscle activation, moderate-to-high-level muscle activation, and high-to-very-high muscle activation of the RA and RF could be achieved in Prone_con_, Prone-Feet_suspension_, and Prone-Arm_suspension_, respectively.

In supine bridge exercises, the anterior muscles (RA and RF) remained in the low activation level, whereas their muscle activities were significantly higher in Supine-Feet_suspension_ than in Supine_con_. The addition of the destabilizing components might increase perturbation of the trunk. Thus, the demand of coactivation of the global and local core musculature was increased to produce comparable counterforces to offset the rotational torque and maintain the same pelvic level on both sides. Significantly higher activation of the BF was observed in Feet_suspension_ than in Supine_con_. Considering that the long head of BF muscle was attached to the ischial tuberosity, which was responsible for posterior pelvic rotation, high demand of RA and RF for generating the torque in the opposite direction was required to counteract any undesirable posterior pelvic rotation induced by the BF. When comparing our results regarding the RA activity in supine bridge conditions with those of recent studies, conflicting findings were observed. One of the studies revealed that RA activation was significantly higher in Supine-Feet_suspension_ than in Supine_con_ [20]. On the contrary, another finding indicated that RA activation remained unchanged between foot-suspended and stable conditions [22]. Our findings are congruent with the former results. Subjects performed supine bridge exercises in the study conducted by Calatayud et al. (2017) using a suspension system with straps differently designed, which required participants to place the sole of their feet flat on the strap. Consequently, the researchers hypothesized that the biomechanical characteristics between Supine-Feet_suspension_ and Supine_con_ exercises were similar. However, the suspension straps used in the current study and in the study conducted by Harris et al. (2017) required subjects to anchor their heels on the straps. With this setting, the researchers hypothesized that high forward sliding moment potentially increased the isometric works on the BF directly and the RA indirectly for stabilization purposes.

Regarding the activation of the GM in our study, the significant decrease of its activities in Supine-Feet_suspension_ deviated from the previous findings reporting no significance in its activities across Supine_con_ and Supine-Feet_suspension_ [20]. Although a significant difference in GM activity was observed among supine bridge exercises in this study, only moderate activity level was found in Supine_con_ and Supine-Arm_suspension_. The supine bridge exercises only induced the moderate muscle activity level on the GM.

The lumbopelvic complex played an important role in transferring forces through body segments and kinetic chains. The %MVC was determined by the ratio between the resistance and muscle mass [44]. During bridge exercises, the contributing ratio between the upper and lower anterior musculature (RA and RF) and posterior musculature (TES, LM, and BF) in the lumbopelvic complex changed along the shift of the center of mass under different suspension conditions and the adjustment of the direction of force exerted on body segments because of the change of relative body position on the base of support. Supine-Feet_suspension_ was the most challenging suspension condition because of the increased strap pulling force on knee extension. Maintaining the body form under this body position required an intense sustained knee flexion, which could be demonstrated by the very high activation level of the BF in this study. This finding also indicated the high activation level of TES and LM in stabilizing the pelvis to boost the efficiency of force transmission through the kinetic chain under Supine-Feet_suspension_. In line with our current findings, the same study [22] also pointed out a significantly greater TES activation under Supine-Feet_suspension_. Therefore, Supine-Feet_suspension_ is a good option when compared with Supine_con_ and Supine-Arm_suspension_ in achieving high level of LM and TES muscle activities and very high level of BF muscle activity.

The rectified and smoothed EMG signal was positively related to the amount of force produced by the muscle; hence, it could provide a general guideline as to the difficulty of the exercise [45]. Loads of 45–50% of one repetition at maximum effort (1RM) have been shown to increase strength in previously untrained individuals [46,47,48]. The addition of the suspension training system could effectively increase the anterior and posterior core muscle activation from low level to very high level, thereby improving core training exercises in a similar body position for strengthening purposes. Furthermore, bridge exercises with suspension were ideal transitional training for athletes to improve strengthening exercises on an unstable base of support. Literature findings indicated that the RA activation level when performing dynamic exercises, such as push-up, in suspension was very high (88%), but it was moderate (24%) when such exercises were performed on the floor [49]. The isometric nature of suspended bridge exercises could fulfill the prerequisite of high RA activation in advanced dynamic strengthening exercises. The implicit instability of suspension training could also be utilized to train core-stabilizing musculature, and its specificity could effectively mimic the loads of particular athletes such as swimmers [11].

## 5. Practical Applications and Study Limitations

This study found that the RA, RF, TES, LM, and BF had higher muscle activities in Supine-Feet_suspension_ than in Supine-Arm_suspension_ and Supine_con_. The activation of the RF, TES, and BF was higher under Supine-Arm_suspension_ as compared with Supine_con_. Supine-Feet_suspension_ elicited significantly lower muscle activity for GM when compared with Supine-Arm_suspension_ and Supine_con_. Therefore, if the target exercise muscle groups were the RA and RF, then Prone-Arm_suspension_ was recommended. However, if the target exercise muscle groups were the TES, LM, and BF, then Supine-Feet_suspension_ was recommended. Such findings might be important for health professionals or strength and conditioning coaches to confirm the use of placing different body parts on suspension straps for core stability improvement or specific muscular training.

The acute responses of core muscle activity during bridge exercises were identified in this study. In a more practical situation, the change of muscle activity in prolonged bridge exercises may be another research area because some people may perform bridge exercises for several minutes.

## Figures and Tables

**Figure 1 ijerph-18-05908-f001:**
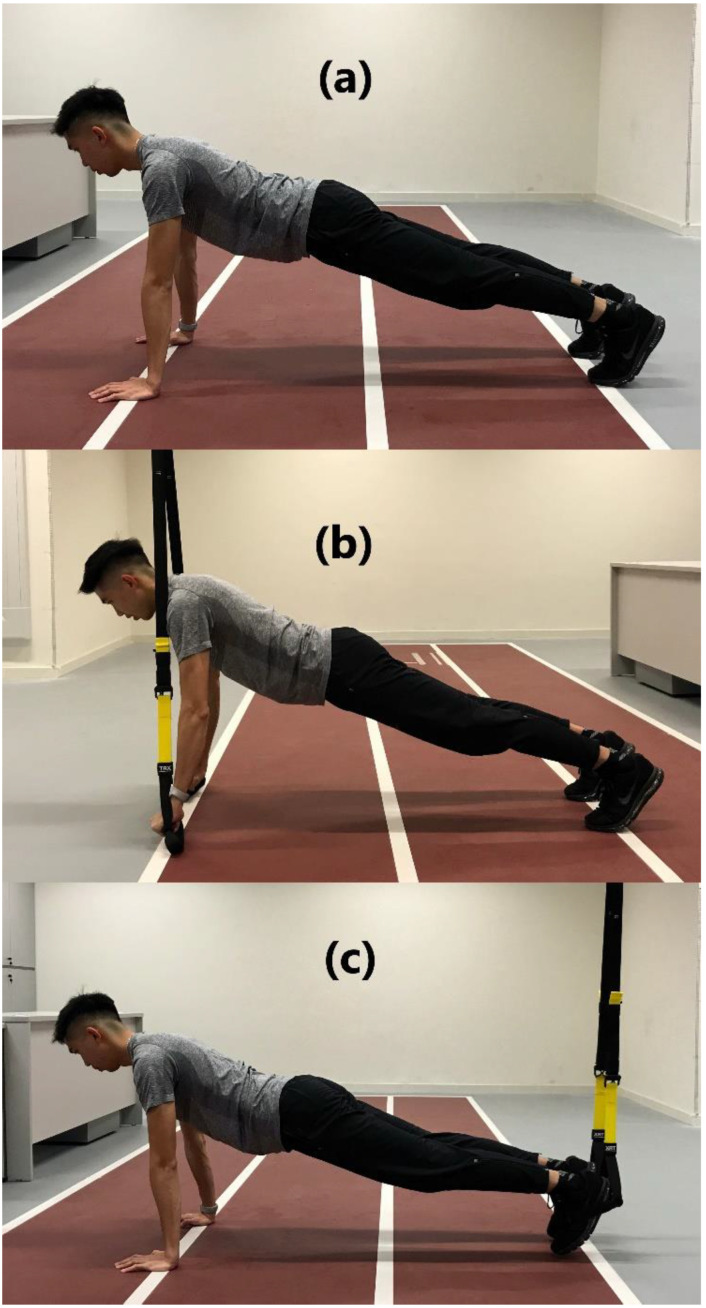
(**a**) Regular prone bridge (Prone_con_) is a prone bridge position on an exercise mat with arms held perpendicular to ground level. Only the palms and toes were in contact with the ground. Subjects were instructed to maintain a neutral head and spine with extended leg position throughout the exercise; (**b**) prone bridge with arm suspension (Prone-Arm_suspension_) is similar to the regular prone bridge position with the feet placed together on the ground, while the arms were placed inside the suspension straps with a neutral grip position and straight-arm position perpendicular to ground level; (**c**) prone bridge with feet placed on a suspension system (Prone-Feet_suspension_) is similar to the regular prone bridge position with palms placed on the ground, while the instep was placed on the suspension straps.

**Figure 2 ijerph-18-05908-f002:**
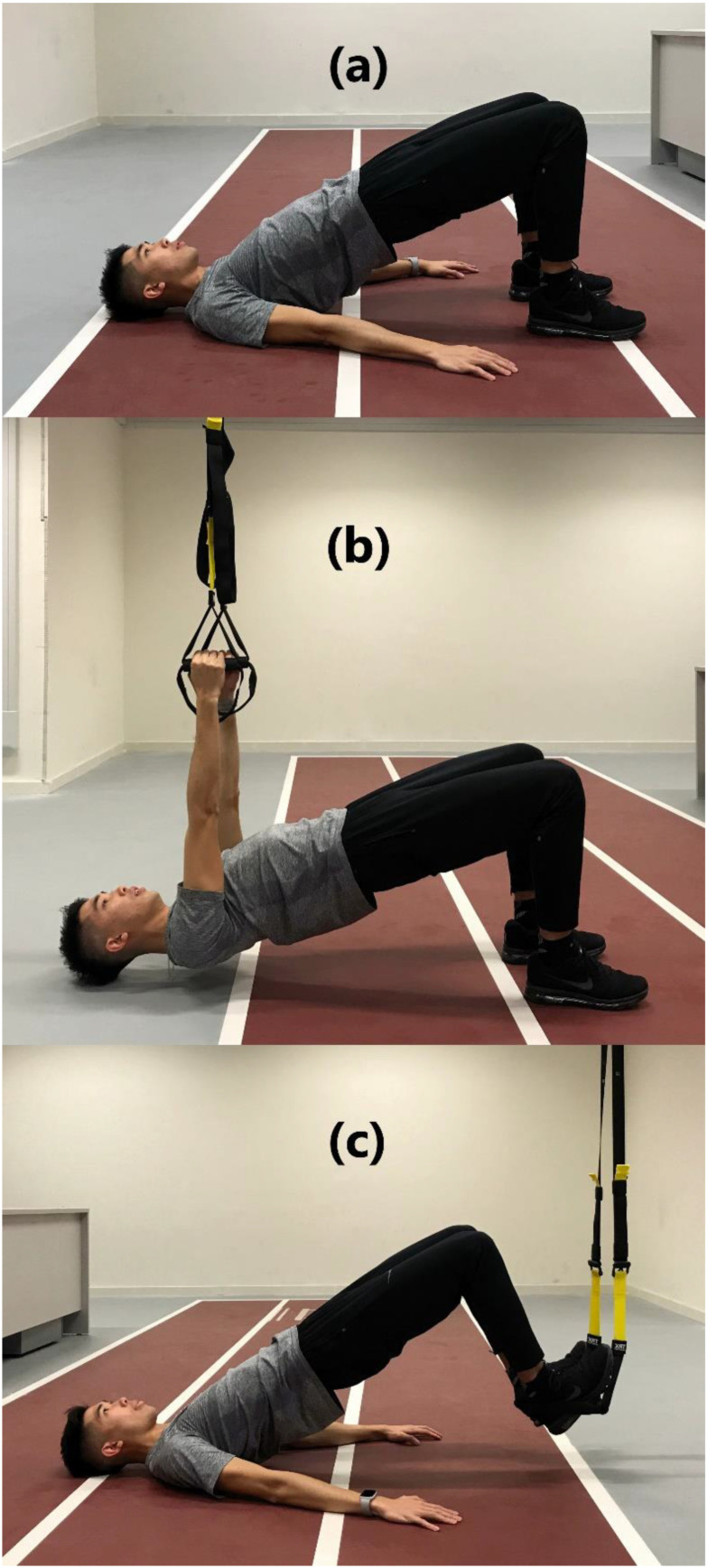
(**a**) Regular supine bridge (Supine_con_) is a supine bridge position with feet shoulder-width apart. The arms were placed beside the torso, and the knees were flexed at 90° with both feet resting on the exercise mat. The pelvis was lifted and aligned with the thigh; (**b**) supine bridge with arm suspension (Supine-Arm_suspension_) is similar to the regular supine bridge position, but the suspension handle was held in a neutral grip position and straight-arm position being perpendicular to ground level; (**c**) supine bridge with feet on a suspension system (Supine-Feet_suspension_) is similar to the regular supine bridge position, but the feet were placed onto the suspension strap in shoulder width.

**Figure 3 ijerph-18-05908-f003:**
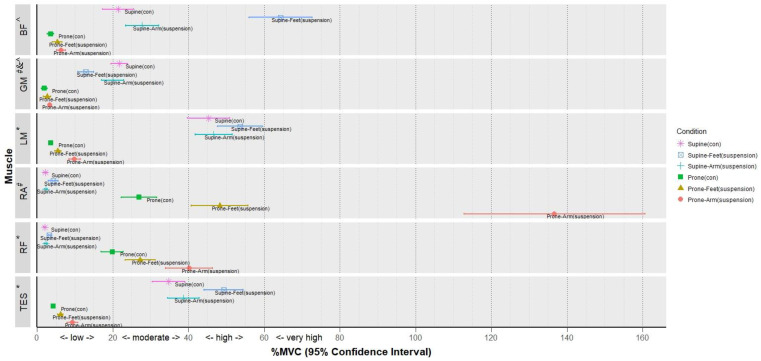
Muscle activity (%MVC; mean and 95% CI) during six bridge exercises. ***Notes:*** Bicep femoris (BF), gluteus maximus (GM), lumbar multifidus (LM), rectus abdominis (RA), rectus femoris (RF), thoracic erector spinae (TES), regular prone bridge (Prone_con_), prone bridge with arms suspension (Prone-Arm_suspsension_), prone bridge with feet suspension (Prone-Feet_suspsension_), regular supine bridge (Supine_con_), supine bridge with arms suspension (Supine-Arm_suspsension_), and supine bridge with feet suspension (Supine-Feet_suspsension_). **p* < 0.01 among all exercises; ^#^*p* < 0.01 among most exercises except for the comparison between Supine_con_ and Supine-Arm_suspsension_; ^^^*p* < 0.01 among most exercises except for the comparison between Prone-Feet_suspsension_ and Prone-Arm_suspsension_; ^#&^^*p* < 0.01 among most exercises except comparison between Prone-Feet_suspsension_ and Prone_con_.

**Table 1 ijerph-18-05908-t001:** Test–retest reliability of sEMG recordings (after MVC normalization) within the same day.

Variations	Prone_con_	Prone-Feet_suspension_	Prone-Arm_suspension_	Supine_con_	Supine-Feet_suspension_	Supine-Arm_suspension_
RA	0.8	0.81	0.97	0.84	0.94	0.91
RF	0.87	0.85	0.9	0.86	0.9	0.87
TES	0.9	0.75	0.91	0.94	0.89	0.88
LM	0.89	0.8	0.69	0.88	0.92	0.77
GM	0.82	0.56	0.75	0.81	0.89	0.92
BF	0.93	0.87	0.8	0.81	0.9	0.86

Note: Values are intraclass correlation coefficient (ICC). 0.50–0.75: moderate reliability, orange 0.76–0.90: good reliability, light green >0.90: excellent reliability, dark green. Rectus abdominis (RA), rectus femoris (RF), thoracic erector spinae (TES), lumbar multifidus (LM), gluteus maximus (GM), biceps femoris (BF), regular prone bridge (Prone_con_), prone bridge with arm suspension (Prone-Arm_suspension_), prone bridge with feet on a suspension system (Prone-Feet_suspension_), regular supine bridge (Supine_con_), supine bridge with arm suspension (Supine-Arm_suspension_), and supine bridge with feet on a suspension system (Supine-Feet_suspension_).

**Table 2 ijerph-18-05908-t002:** Muscle activity (%MVC) during isometric bridge exercises under six different conditions.

Muscle	Prone_con_		Prone-Feet_suspension_		Prone-Arm_suspension_		Supine_con_		Supine-Feet_suspension_		Supine-Arm_suspension_	
	Mean ± SD	95% CI	Mean ± SD	95% CI	Mean ± SD	95% CI	Mean ± SD	95% CI	Mean ± SD	95% CI	Mean ± SD	95% CI
RA ^#^	26.9 ± 15.1	22.3, 31.6	48.3 ± 24.3	40.8, 55.7	136.7 ± 77.7	112.8, 160.6	2.2 ± 1.8	1.6, 2.7	4.3 ± 4.2	3.0, 5.6	2.4 ± 1.3	2.0, 2.7
RF *	19.9 ± 9.5	17.0, 22.8	27.2 ± 12.9	23.2, 31.2	40.2 ± 20.1	34.0, 46.4	2.0 ± 1.1	1.7, 2.4	3.2 ± 1.8	2.6, 3.7	2.3 ± 1.3	1.9, 2.7
TES *	4.2 ± 1.5	3.8, 4.7	6.2 ± 2.4	5.4, 6.9	9.3 ± 4.7	7.9, 10.7	34.8 ± 14.1	30.4, 39.1	49.3 ± 16.7	44.2, 54.5	38.7 ± 13.6	34.5, 42.9
LM *	3.6 ± 1.9	3.0, 4.2	5.5 ± 2.6	4.7, 6.3	9.9 ± 5.0	8.4, 11.5	45.3 ± 18.3	39.7, 51.0	53.6 ± 19.6	47.6, 59.6	46.7 ± 15.9	41.8, 51.6
GM ^&^#^	1.9 ± 2.7	1.1, 2.7	2.6 ± 3.0	1.7, 3.6	3.3 ± 1.6	2.9, 3.8	21.7 ± 7.3	19.5, 24.0	12.9 ± 6.7	10.8, 14.9	20.0 ± 9.6	17.0, 23.0
BF ^^^	3.6 ± 2.8	2.7, 4.4	5.4 ± 4.3	4.0, 6.7	6.3 ± 4.0	5.1, 7.6	21.4 ± 13.2	17.3, 25.5	64.3 ± 27.2	56.0, 72.7	27.8 ± 14.1	23.4, 32.1

**Note:** Rectus abdominis (RA), rectus femoris (RF), thoracic erector spinae (TES), lumbar multifidus (LM), gluteus maximus (GM), biceps femoris (BF), regular prone bridge (Prone_con_), prone bridge with arm suspension (Prone-Arm_suspension_), prone bridge with feet on a suspension system (Prone-Feet_suspension_), regular supine bridge (Supine_con_), supine bridge with arm suspension (Supine-Arm_suspension_), and supine bridge with feet on a suspension system (Supine-Feet_suspension_). * *p* < 0.010 among all exercises; ^#^
*p* < 0.010 among most exercises except for the comparison between Supine_con_ and Supine-Arm_suspension_; ^^^
*p* < 0.010 among most exercises except for the comparison between Prone-Feet_suspension_ and Prone-Arm_suspension_; ^&^#^
*p* < 0.010 among most exercises except for the comparison between Prone-Feet_suspension_ and Prone_con_.

## Data Availability

The data presented in this study are available on request from the corresponding author.

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
