# Peer review of "Acute Responses of Core Muscle Activity during Bridge Exercises on the Floor vs. the Suspension System"

_ijerph, 2021, doi:10.3390/ijerph18115908_

Round 1
Reviewer 1 Report
First of all, congratulations to the authors for this research article.
1) The abstract must be unstructured, so remove the parts of background, methods ...
2) I believe that it is absolutely a priority to include the previous experience or the background in training of the sample, since the muscular electrical activity and the adequacy and response to different exercises will vary enormously depending on the experience and previous interindividual training of the subjects.
3) Figure 3, I understand that it is figure 1 since there is no more in the document, likewise I do not know if it will be due to the platform when editing the document to PDF, but this is illegible ... (maybe its my old comp), just check in case pls.
4) Regarding table 3, eliminate unnecessary spaces.
5) Finally, include a section on practical applications and study limitations.
Author Response
Dear Editor and Reviewers,
Attached please find a revised manuscript by Jim T.C. LUK, Freeman K.C. KWOK, Indy M.K. HO, Del P. WONG. The title is “Acute responses of core muscle activity during bridge exercises on the floor vs. the suspension system”.
I would express my sincere thanks for the comments and suggestions on my manuscript, which offered a lot of improvement and make significant contribution in the academic area. Please find my following replies for your further considerations.
1) The abstract must be unstructured, so remove the parts of background, methods ...
The abstract format has been revised according to the suggestion from reviewer.
2) I believe that it is absolutely a priority to include the previous experience or the background in training of the sample, since the muscular electrical activity and the adequacy and response to different exercises will vary enormously depending on the experience and previous interindividual training of the subjects.
Yes, we have asked the participant about their previous experience or background of training with suspension system as most of them are our sports programme students. The inclusion criteria No. 5 was added after revision. (Line 99-100, under simple Markup view of Track Changes function in WORD)
3) Figure 3, I understand that it is figure 1 since there is no more in the document, likewise I do not know if it will be due to the platform when editing the document to PDF, but this is illegible ... (maybe its my old comp), just check in case pls.
Totally 3 figures were clearly display within the document.
4) Regarding table 3, eliminate unnecessary spaces.
Table 3 was removed, the same data presentation could be found in Figure 3 indeed.
5) Finally, include a section on practical applications and study limitations.
The section “Practical applications and study limitations” was added at Line 107 and the study limitation was shown at Line 118-121.

Reviewer 2 Report
I believe that the topic of the study should be of interest to the readers in the International Journal of Environmental Research and Public Health. The present manuscript is an applied article with regard to exercise performance and recovery. Several mistakes or unclear parts have been identified. The authors must interpret their findings in the context of supporting/not supporting existing literature and provide acute responses of core muscle activity applicability. However, I believe the authors need to revise some of the contents of the paper.
- Please display the three digits after the decimal point uniformly for the P value.
- The findings of this study are similar to many previous studies. What is the biggest research highlight of this study?
- What is the difference between Table 3 and Figure 3? What do you want to express?
- The paper requires proofreading as there are many problems with syntax and use of punctuation that need to be addressed.
- The use of paragraphs made up of a single sentence is pervasive and should be corrected throughout the manuscript.
Author Response
Dear Editor and Reviewers,
Attached please find a revised manuscript by Jim T.C. LUK, Freeman K.C. KWOK, Indy M.K. HO, Del P. WONG. The title is “Acute responses of core muscle activity during bridge exercises on the floor vs. the suspension system”.
I would express my sincere thanks for the comments and suggestions on my manuscript, which offered a lot of improvement and make significant contribution in the academic area. Please find my following replies for your further considerations.
- Please display the three digits after the decimal point uniformly for the P value.
The decimal point of all P values were revised according to the comments of reviewer.
- The findings of this study are similar to many previous studies. What is the biggest research highlight of this study?
The highlight was added at Line 68-71, “The comparison of muscle activities between stable and unstable surfaces is better with similar posture and trunk inclination; however, most of the literature compares the exercises with very different trunk inclination because of the use of swiss balls or improper length of suspension straps [11,26,27]”
- The paper requires proofreading as there are many problems with syntax and use of punctuation that need to be addressed.
- The use of paragraphs made up of a single sentence is pervasive and should be corrected throughout the manuscript.
Authors have followed the suggestions from reviewers and editor for the extensive English revisions and editing services which checked by a native English-speaking colleague.
